# Construction of VAE-GRU-XGBoost intrusion detection model for network security

Yu Chen, Xiaohong Zheng *, Nan Wang

Zhangjiakou Open University, Zhangjiakou, China

* zhengxiaoh882024@163.com

## Abstract

With the advent of the big data era, the threat of network security is becoming increasingly severe. In order to cope with complex network attacks and ensure network security, a network intrusion detection model is constructed relying on deep learning technology. In order to extract and analyze network intrusion features, this study uses variational auto-encoders to extract and reduce the dimensionality of the invaded network traffic, and combines the advantages of extreme gradient boosting to perform classification tasks. Finally, a network intrusion detection model for network security is constructed by combining the gated recurrent unit. The results showed the area under the curve of the research model reached 97.48% and 95.24% in the KDD99 dataset and OODS dataset, respectively. In the confusion matrix experiment, the model achieved classification accuracy greater than 0.91 for different attack traffic samples in both the training and testing sets. When the sample sizes were 10000 and 40000, the shortest time and longest feature extraction time of the model were 0.030s and 0.112s, respectively. In summary, the constructed model on the basis of improved variational auto-encoder for network security has high accuracy in network intrusion detection.

## 1. Introduction

As artificial intelligence continues to evolve, the frequency and complexity of network attacks are also increasing. Intrusion detection technology is a vital technique for detecting potential security threats in network systems, which detects abnormal information by analyzing network traffic data to identify possible intrusion behaviors [1]. Traditional intrusion detection methods typically rely on signature detection, which mainly relies on machine learning algorithms and manual feature extraction [2]. However, the current data information presents complex features such as non-linearity, imbalanced distribution, and high dimensionality, making it difficult for traditional intrusion detection methods to effectively respond to new and unknown network attack methods [3]. In addition, the multi-class imbalance, outlier points, and

**Data availability statement:** All relevant data are within the manuscript and its Supporting Information files.

**Funding:** The author(s) received no specific funding for this work.

**Competing interests:** The authors have declared that no competing interests exist.

concept drift problems that occur in network data streams can greatly affect prediction performance. Most existing classification algorithms cannot learn with a small number of sample labels [4]. In this context, deep learning technology has become an effective tool in network intrusion detection due to its powerful data processing capabilities. Variational Auto-Encoder (VAE) in deep learning technology has the characteristic of extracting potential features from data. Gated Recurrent Unit (GRU) is an improved Recurrent Neural Network (RNN), which performs well in processing time series data [5]. Extreme Gradient Boosting (XGBoost) is an ensemble learning method, which has efficient classification ability [6]. Therefore, a hybrid model integrating VAE, GRU, and XGBoost is innovatively constructed. This model utilizes VAE to capture potential features of network traffic data. The GRU is applied to process time series data, deeply mining the potential features of the data. Finally, by combining the classification advantages of XGBoost, the detection accuracy is further improved. The research content mainly includes four parts. The second part reviews the current research status of intrusion detection and deep learning technology both domestically and internationally. The third part designs a detection model, in which the first section improves the VAE generation model using GRU to achieve network feature extraction. The second section constructs an intrusion detection model on the basis of VAE-GRU-XGBoost. The fourth part is to validate the VAE-GRU-XGBoost.

## 2. Related works

As the Internet continues to grow and evolve, network security has become increasingly crucial, so network intrusion detection has attracted much attention. Fu Y et al. used deep learning model and bidirectional long short-term memory networks to construct a traffic anomaly detection model in response to increasingly complex network attack methods. This model used Convolutional Neural Network (CNN) to extract sequence features of data traffic and utilized attention mechanisms for weight allocation. The detection accuracy was as high as 90.73% [7]. In order to ensure network security, He K et al. took deep neural network architecture to construct a detection system. In response to the limitation of low classification accuracy caused by adversarial attacks on deep neural network architecture, computer vision technology was introduced to improve the system. The results showed that the improved system could effectively resist white box and black box adversarial attacks, effectively ensuring network security [8]. Khan M A et al. used deep learning and machine learning technology to construct an intrusion system in order to automatically detect malicious threats. The system used CNN to extract local features. The convolutional RNN structure was used to classify and predict malicious network attacks. The system was experimentally verified on a dataset. The detection accuracy for high malicious attacks was as high as 97.75% [9]. Alzahrani A O et al. used machine learning method to construct an intrusion detection system for real-time monitoring of malicious behavior in the network. The system utilized random forests to monitor traffic and XGBoost to classify attack types. The system had an accuracy rate of 95.95% in classifying network attacks, demonstrating its effectiveness [10]. Injadat M N et al. used machine learning to design a detection method. The system framework was

based on multi-stage optimization of machine learning and used sampling techniques to determine the minimum training sample size. The system had a minimum false positive rate of 1.2% for network attack recognition, and had high recognition accuracy [11].

Deep learning technology exerts a crucial function in the design of network intrusion detection systems. Song Y et al. took deep learning technology to construct an intelligent network intrusion detection system to optimize the attack detection accuracy. The auto-encoder was used to classify unknown attack types. This method used a benchmark dataset to train the encoder and find the optimal hyper-parameter settings for the auto-encoder. This method was feasible [12]. Qiu H et al. utilized deep learning technology to design a network security protection model in response to new types of network attack methods. The model extracted massive training data to replicate the black box model and used significance maps to reveal the key features of each packet attribute. The model could accurately identify 94.31% of attack success rates [13]. Cao B et al. combined deep learning technology and GRU design to improve the intrusion detection accuracy. The model utilized fused CNN to extract spatial features, random forest algorithm for feature screening, and softmax function for classification. This method effectively solved the low classification accuracy [14]. Atefinia R et al. used deep learning techniques to detect attacks on signatures in network traffic in response to the expanding network attacks. This method utilized deep neural network models to construct a multi-modular intrusion detection system, and set different weights for different modules to reduce false alarm rates. This method effectively improved the recognition rate of network attacks [15]. Almomani O et al. proposed a deep learning technology combined with data mining for anomaly detection to identify malicious behavior in the network. This method utilized optimized particle swarm optimization algorithm to capture a large number of temporal features. Then, the Support Vector Machine (SVM) was applied to select the features. This method effectively improved the recognition accuracy [16].

In summary, although there have been many achievements in research on network intrusion detection and deep learning, there are still relatively few studies that combine the advantages of various models such as auto-encoders, XGBoost, and GRU for network intrusion detection. For example, Cao B et al.'s model extracts spatial features through convolutional neural networks (CNN) and then uses softmax for classification. Although there are some similarities between this study and the model presented in this article, there are differences in their methods. The model in this article uses GRU to capture temporal features and adopts XGBoost classifier, emphasizing stronger classification ability in complex temporal data and high-dimensional features. Therefore, the research innovatively integrates the advantages of VAE, GRU, and XGBoost to construct a VAE-GRU-XGBoost network intrusion detection model that is oriented towards network security to optimize the detection accuracy.

## 3. Design of network intrusion detection model based on VAE-GRU-XGBoost

This study combines the advantages of VAE and GRU to design an improved VAE feature extraction model based on GRU. Combined with XGBoost classifier, an intrusion detection model on the basis of VAE-GRU-XGBoost is designed to detect abnormal network traffic.

### 3.1. VAE feature extraction model based on GRU

As the Internet continues to evolve at a rapid pace, intelligent informatization not only provides great convenience for people, but also brings more network attacks. Traditional network intrusion anomaly detection methods have limitations such as low computational efficiency, strong data sparsity, and difficulty in defining in different scenarios [17]. As a generative model, VAE can identify abnormal behavior by learning the potential distribution of data [18]. The core idea is to map input data to latent space and measure whether the data conforms to a normal distribution by reconstructing errors [19]. Compared with traditional auto-encoders, VAE introduces probabilistic models, making it more advantageous in handling complex data. Accordingly, the VAE is applied for anomaly detection. In order to improve the accuracy of network intrusion detection, three technologies, GRU, VAE, and XGBoost, were studied and combined. In the model design, GRU

 

was used to capture temporal features, VAE was used for latent feature extraction, and XGBoost was used as a classifier for accurate classification. The reason for choosing this specific combination is that GRU can effectively capture time series features in the input data, avoiding the gradient vanishing problem in traditional RNNs. GRU has lower computational complexity compared to traditional LSTM and RNN when dealing with long-term dependencies, thus improving the real-time performance of network intrusion detection. VAE can extract potential feature representations from data through variational autoencoders, especially on high-dimensional data, where VAE has strong feature representation capabilities. As an ensemble learning algorithm based on decision trees, XGBoost can perform accurate classification on existing features, especially suitable for classification tasks of high-dimensional data. The VAE is displayed in Fig 1.

In Fig 1, the VAE is based on auto-encoder, which is divided into three layers: encoder, intermediate layer, and decoder. Unlike auto-encoders, the middle layer of VAE is also known as the sampling layer, which calculates the latent vector of the sample by computing the Gaussian distribution. This feature enables VAE to generate output data that conforms to the latent distribution, and its unsupervised learning characteristics also make it commonly used in anomaly detection. In VAE, the encoder maps the input data to the latent variable space. The sampling layer utilizes reparameterization techniques to convert random noise into latent variables. The decoder maps the latent variables back to the original data space, producing samples that are similar in distribution to the input data. In network intrusion detection tasks, VAE extracts specific features such as connection duration, packet size and frequency, protocol type, connection method, and packet content from raw network traffic. One of the important goals of VAE in feature extraction and dimensionality reduction is to ensure that key information in the original data is preserved as much as possible while reducing dimensionality. The reconstruction error can be determined by comparing the difference between the original input data and the data reconstructed by VAE through the decoder to determine whether dimensionality reduction has caused the loss of key information. Low reconstruction error means that VAE can effectively preserve the key information of the input data. The encoder consists of multiple hidden layers, with the goal of mapping the input to latent space. The output of the encoder is the latent variable $z$, which represents the distribution of input data in the latent space. The mathematical formula of VAE encoder is shown in equation (1)

$$q\left(z \mid x\right) = \mathbb{N}\left(z \mid u(x), \sigma^2(x)\right)$$

(1)

In equation (1), $u(x)$ and $\sigma^2(x)$ represent the mean and standard deviation, respectively. The encoder maps the input $x$ to latent space through a series of fully connected layers. The decoder maps the representation $z$ of the latent space back to the original data space and reconstructs the data $\widehat{x}$. The decoder is also composed of multiple fully connected layers, and its output is the mean $\widehat{x}$ of the reconstructed data [20]. The mathematical expression for the decoding process is shown in equation (2).

$$p\left(x \mid z\right) = \mathbb{N}\left(x \mid \widehat{x}(z), \sigma^2\right)$$

(2)

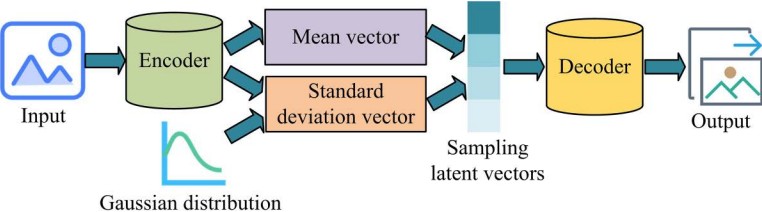

Gaussian distribution

**Fig 1. The basic structure of VAE.**

In equation (2), $\widehat{x}\,(z)$ represents the reconstructed data generated by the decoder. The difference between data can be measured by reconstruction error, which is expressed as equation (3).

$$D(p||q) = \mathbb{E}_{x\sim h}(\log p(x) - \log q(h))$$

(3)

In equation (3), $D$ represents the reconstruction error. $p\,(x)$ represents the probability distribution of the input data. $q\,(h)$ represents the probability distribution of the encoded output data. The Gaussian distribution expression is shown in equation (4).

$$p(h|x) = \frac{p(x|h)p(h)}{p(x)}$$

(4)

In equation (4), $p\,(h|x)$ represents Gaussian distribution. Based on equations (3) and (4), the basic formula for VAE can be obtained, as shown in equation (5).

$$\log p(x) - D[q(h|x)||p(h|x)] = \mathbb{E}_{x\sim h}[\log p(x|h)] - D_{KL}[q(h|x)||p(h)]$$

(5)

In equation (5), $\log p(x) - D[q(h|x)||p(h|x)]$ represents the possibility of VAE reconstruction. $\mathbb{E}_{x\sim h}[\log p(x|h)] - D_{KL}[q(h|x)||p(h)]$ is used to verify the similarity between the true distribution and the approximate distribution [21]. Although VAE can effectively capture data features, it has limitations in explaining the extracted features. Due to its gating mechanism, GRU can better capture long-term dependencies and has the advantage of reducing gradient vanishing problems [22]. Therefore, the study introduces GRU into the VAE structure to extract the intrinsic structure of network traffic data by constraining reconstruction errors and sparse losses. The GRU and VAE parts are responsible for extracting and encoding data features, while XGBoost serves as a classifier to optimize the extracted features. Although end-to-end training is not possible, this design can classify through effective ensemble methods after feature extraction, which can improve the accuracy and stability of the model. The auto-encoder structure of GRU is displayed in Fig 2.

In Fig 2, the GRU auto-encoder combines the advantages of both GRU and auto-encoder. GRU is capable of processing sequential data and controlling the flow of information by introducing update and reset gates, thereby better capturing long-term dependencies in the sequence. The encoder and decoder structures of auto-encoders can reconstruct the original input data. In this structure, the input of the auto-encoder is the hidden state of GRU. The output is the original data that is attempted to be reconstructed. In this way, the structure can maintain the main features of the data, and also reduce the data dimensionality, thereby simplifying subsequent prediction tasks. The hidden state of GRU is shown in equation (6).

$$(h_1, h_2, \cdots, h_T) = GRU_{encoder}(P)$$

(6)

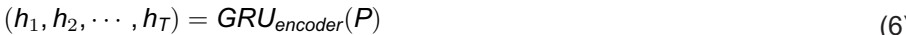
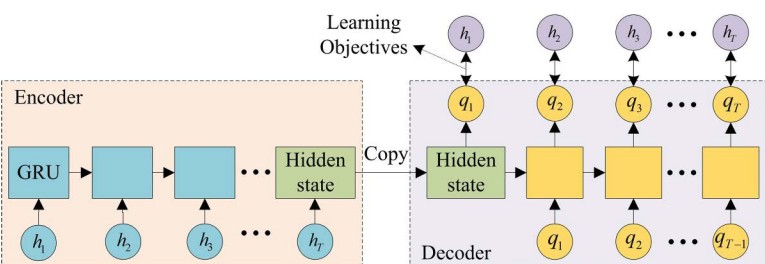

**Fig 2. The auto-encoder structure of GRU.**

In equation (6), $h_T$ represents the hidden state of GRU. $T$ represents the quantity of input data [23]. The reconstructed sequence of GRU decoding is shown in equation (7).

$$(q_1, q_2, \cdots, q_T) = GRU_{decoder}(h_T)$$

(7)

In equation (7), $q_T$ represents the reconstructed sequence. GRU, as a variant of recurrent neural network, can capture temporal features, extract temporal relationships of potential features, and enhance dynamic feature learning in network intrusion models. In addition, GRU has a simpler structure and fewer parameters compared to commonly used LSTM structures, so GRU can reduce the risk of model overfitting during training compared to LSTM. So the study chose GRU to improve the VAE feature extraction model. The VAE feature extraction model based on GRU improvement is shown in Fig 3.

In Fig 3, in the improved VAE on the ground of GRU, the hidden state is initialized. Then, the input data needs to be preprocessed. The preprocessing steps include data cleaning, data conversion, and data sampling. Then, the processed features are input into the encoding area of the model for learning. At this point, it is necessary to extract the universal feature information from the traffic and obtain the corresponding encoding results based on this information. Then, the previous reconstruction value is input into the GRU decoder to generate a new reconstruction value. Finally, the convergence result is obtained by calculating the loss function. Equation (8) displays the loss function.

$$L = q(h|x) \log \int \left( \frac{P(h, x)}{q(h|x)} \right) dh$$

(8)

In equation (8), $L$ represents the loss function. In network intrusion detection, normal traffic data are applied to train the VAE, enabling them to learn the feature distribution of normal traffic and accurately identify abnormal traffic during actual detection [24]. The VAE feature extraction model based on GRU improvement can automatically extract potential features of data, avoiding the complex manual feature extraction and improving the efficiency and accuracy of feature extraction. The purpose of applying VAE in the GRU model process is to leverage the advantages of GRU in processing sequential data, while utilizing the potential space and generation capabilities of VAE to better extract and model the features of network traffic. The encoder of VAE is responsible for mapping input data to latent space, while GRU models temporal data through its gating mechanism, effectively capturing long-term dependencies in the time series during the encoding process. Therefore, the combination of VAE and GRU can not only effectively detect anomalies, but also automatically extract potential features from network traffic, avoiding the complexity of traditional manual feature extraction and improving the efficiency and accuracy of feature extraction. The application process of VAE in GRU model is as follows: (1) The encoder

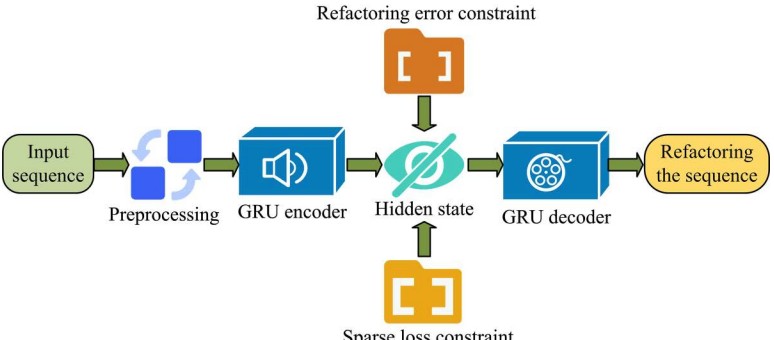

**Fig 3. VAE feature extraction model based on GRU improvement.**

part of VAE receives the original network traffic data and maps it to latent space. (2) After the VAE encoder maps the data to latent space, GRU uses these latent variables to model the long-term dependencies of time series data. (3) GRU can better understand the evolution patterns of time-series data by gradually processing input data and combining latent variables. (4) After GRU completes the temporal modeling, the decoder part of VAE can convert latent variables back to the original data space, generating a reconstruction of the input data. (5) Finally, by calculating the reconstruction error, it can be determined whether there is an anomaly.

### 3.2. Network intrusion detection model based on VAE-GRU-XGBoost

GRU is capable of capturing time series features of data. After effectively extracting the potential features of the data using VAE's feature extraction capability, a powerful classifier is required to classify the types of network intrusions. As a machine learning algorithm, XGBoost can accurately classify and judge based on the extracted features [25]. Therefore, based on the improved VAE feature extraction model optimized by GRU, a comprehensive network intrusion detection model is constructed by combining XGBoost classifier to adapt to complex and changing network attack scenarios. The feature classification process based on XGBoost is shown in Fig 4.

As shown in Fig 4, in the feature classification process based on XGBoost, the input data features are first calculated and sorted according to their importance. Next, the most important feature values are selected and their Area Under Curve (AUC) values are calculated. Then, feature values are added one by one in order of their importance for judgment. If the increase in AUC value is below the threshold, the feature value will be removed. Otherwise, the feature value will be retained and included in the initial dataset of subsequent models. The above process is repeated until all features are calculated and judged, ultimately forming a classified feature subset. The importance of features is determined by calculating the contribution of each feature in the decision tree. Features with higher importance usually provide more valuable information, thus having a greater impact on the classification ability of the model. In order to reduce computational complexity and improve classifier performance, the study adopted a method of adding features in order of their importance. Since the importance of features is calculated based on their contribution to predicting the target variable during model training, theoretically, the more important features have a greater impact on performance improvement. Therefore, the study did not conduct experiments on all possible feature addition orders, but instead determined the order of feature addition based on the calculated feature importance ranking to optimize the efficiency and performance of the model. The objective function of XGBoost is displayed in equation (9).

$$OBJ = \sum_{i=1}^{n}[g_i f_t(x_i) + \frac{1}{2}z_i f_t^2(x_i)] + \sum_{i=1}^{t} u(f_i)$$

(9)

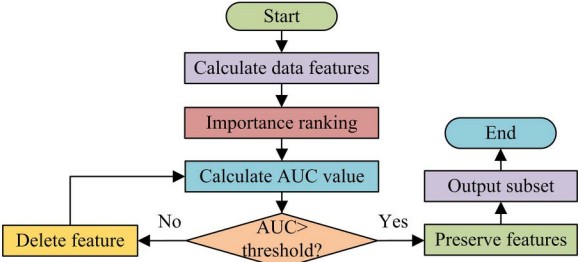

**Fig 4. Feature classification process based on XGBoost.**

In equation ($9$), *OBJ* represents the objective function of XGBoost. $n$ signifies the sample size. $i$ signifies the sample. $g_i$ and $z_i$ represent first-order and second-order derivatives, respectively. $f_t$ represents the $t$-th base learner. $u$ represents the regularization term [26]. The expression of the cross entropy loss function is displayed in equation ($10$).

$$L(y_i, y_i') = y_i \ln(1 + e^{-y_i}) + (1 - y_i') \ln(1 + e^{y'_i}) \tag{10}$$

In equation ($10$), $L(y_i, y_i')$ represents the cross entropy loss function. $e$ represents the natural constant. $y_i$ and $y_i'$ signify the predicted and true values of the $i$-th sample. The expanded first-order and second-order derivatives are shown in equation ($11$).

$$\begin{cases} g_i = \frac{1}{1+e^{-y_i}} - y_i \\ z_i = \frac{1}{1+e^{-y'_i}} * (1 - \frac{1}{1+e^{-y'_i}}) \end{cases} \tag{11}$$

The XGBoost loss function obtained from equations ($10$) and ($11$) is displayed in equation ($12$).

$$L' = \sum_{i=1}^{n} L(y_i, y_i') \tag{12}$$

In equation ($12$), $L'$ signifies the loss function of XGBoost. The Gini index expression of XGBoost is shown in equation ($13$).

$$G_{\text{ini}}(P) = 2P(1 - P) \tag{13}$$

In equation ($13$), $G_{\text{ini}}$ signifies the Gini index. $P$ signifies the probability of having important features [27]. The gain calculation for feature classification using XGBoost is shown in equation ($14$).

$$Gain = \frac{S_L^2}{n_L} + \frac{S_R^2}{n_R} - \frac{S_r^2}{n_r} \tag{14}$$

In equation ($14$), $G_{wn}$ represents gain. $S_L$ and $S_R$ represent the sum of gradients on the left and right sides, respectively. $S_r$ represents the sum of the total gradients. $n_L$ and $n_R$ represent the network traffic data samples on the left and right. $n_r$ represents the total traffic data sample. The mathematical expression for classifying based on gain is shown in equation ($15$).

$$AverageGain = \frac{\sum Gain_x}{FScore} \tag{15}$$

In equation ($15$), *AverageGain* represents the average gain. *FScore* indicates the importance score of the feature [28]. The potential features captured by VAE for different attack modes or new types of network intrusion methods can not only describe existing attack types, but also provide certain generalization ability to deal with new types of attacks. In addition, VAE can also assist in detection by identifying potential feature patterns that differ from normal traffic. As a temporal model, GRU can capture temporal dependencies and dynamic changes in data. For new types of attacks, GRU can respond to the evolution and variation of attack patterns by learning the time-series relationships of potential features. Finally, the XGBoost classifier has learning ability, which can learn the distribution of different types of attacks and recognize new types of attacks through training. In order to evaluate its generalization ability, the study used datasets from

different sources, which contained samples of novel or unknown attack patterns, to verify the model's recognition ability on unseen attack patterns. The steps to evaluate generalization ability are as follows: Firstly, the experiment collects datasets from multiple different sources, so that the dataset not only contains known attack patterns, but also samples of new or unknown attack patterns. This diverse data source ensures the adaptability of the model in the face of different types of attacks. In the data preprocessing stage, feature extraction was performed on the data, using VAE to capture potential features and GRU to analyze time series features. Afterwards, the XGBoost classifier is used to train the extracted features, and the performance of the model on known attack patterns is evaluated through methods such as cross validation, while also focusing on its recognition ability on unknown attack patterns. Finally, the recognition ability of the model is comprehensively evaluated through various performance evaluation indicators such as accuracy and recall rate. The intrusion detection model on the basis of VAE-GRU-XGBoost is shown in Fig 5.

As shown in Fig 5, the VAE-GRU-XGBoost first preprocesses the input dataset, and then inputs it into the improved VAE feature extraction model based on GRU after sampling processing. After extracting potential features, XGBoost is used for classification. In the VAE-GRU-XGBoost, the research first uses GRU encoding structure to reconstruct network features, and uses VAE for feature learning in the case of limited labels. Finally, the XGBoost classification model is applied to classify the data features extracted from the above process to get better detection results. In network intrusion detection, VAE, GRU, and XGBoost are integrated together to form a powerful multi-component model. Firstly, VAE is used for unsupervised feature learning, which maps input data to latent space through an encoder to learn a low dimensional representation of the data. In order to further enhance the model's ability to capture temporal data, GRU is introduced and combined with VAE. The role of GRU is to extract time-dependent features from the input temporal data. Specifically, the input data of the encoder is first processed by a GRU network, which extracts temporal features through GRU's gating mechanism. These temporal features are then fed into VAE for latent spatial mapping, forming a more refined feature representation. Finally, XGBoost serves as a classifier to receive feature representations processed by VAE and GRU, and complete the final network intrusion classification task. XGBoost achieves efficient intrusion detection by learning and classifying these extracted features. In summary, VAE, GRU, and XGBoost work together in this model. VAE is responsible for unsupervised feature learning, GRU enhances temporal feature extraction, and XGBoost completes intrusion classification, forming a collaborative intrusion detection system. The detection process on the basis of VAE-GRU-XGBoost is shown in Fig 6.

As shown in Fig 6, the process contains a training stage and a detection stage. Firstly, the input dataset needs to undergo preprocessing operations such as data cleaning, label transformation, and feature transformation to construct the training and testing sets. In the first stage, the VAE feature extraction model based on GRU improvement is trained on the training set. Then, the training set is input into the trained encoder to obtain feature representations, followed by training the XGBoost classifier. In the detection phase, the testing set inputs the trained encoder to obtain feature representations, and uses the same feature

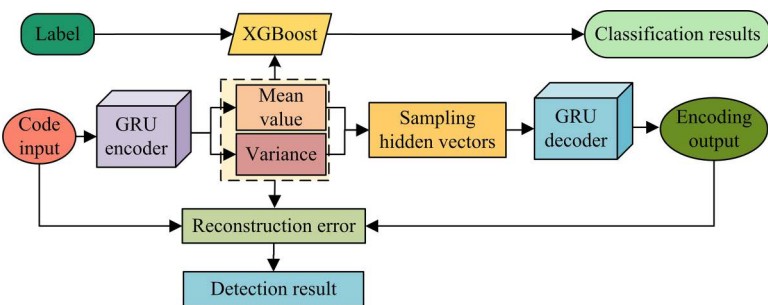

**Fig 5. Intrusion detection model based on VAE-GRU-XGBoost.**

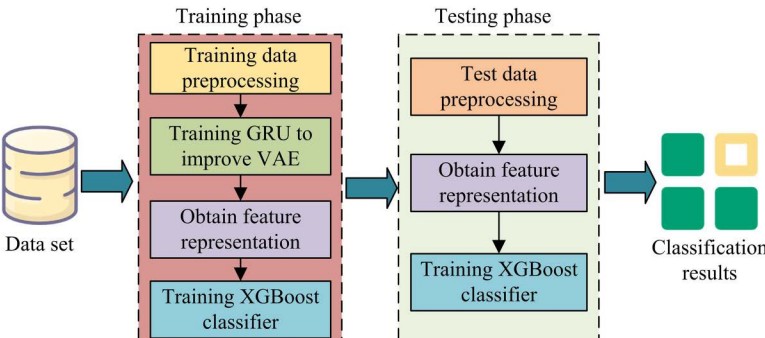

**Fig 6. Intrusion detection process on the basis of VAE-GRU-XGBoost.**

representations to train the XGBoost classifier. Finally, the obtained traffic features are input into the XGBoost classifier to get the final classification result. In the VAE-GRU XGBoost based network intrusion detection model, VAE and GRU are used as deep learning models, while XGBoost is a traditional machine learning model. Due to their different input feature formats, reasonable preprocessing and feature fusion are required to effectively combine these three models and ensure the smooth progress of the experimental process. In data preprocessing, data cleaning is first performed to remove duplicate samples, missing values, and outliers. Then perform label conversion and feature conversion, and convert the labels into standard form through single hot encoding. Afterwards, the combination of VAE and GRU is used for feature extraction. After extracting the features, XGBoost is used as a classifier to classify these features into intrusion types. The key hyperparameters in the VAE-GRU-XGBoost based network intrusion detection model are set as follows: the learning rate of the VAE module is 0.001, the batch size is 64, and the latent dimension is 50; For GRU, the hidden layer dimension is set to 128, the learning rate is set to 0.001, and the Adam optimizer is used for training; For XGBoost, use the default learning rate and tree depth of 6. During the training process, cross validation is used to optimize these hyperparameters to ensure the best performance of the model. In order to improve training efficiency, batch training and early stopping strategies were adopted. During the training process, if the loss function of the validation set does not improve within a certain number of rounds, the training will be stopped.

## 4. Validation of network intrusion detection model based on VAE-GRU-XGBoost

The study first configures a specific experimental environment based on the software and hardware environments, and preprocesses the dataset into a testing set and a training set for subsequent experiments. In the experiment, the performance of the VAE model on the basis of GRU is first verified. Then, the performance of the VAE-GRU-XGBoost model is further validated.

### 4.1. Experimental environment configuration

To validate the VAE-GRU-XGBoost, experiments are carried out on the Ubuntu 18.04 LTS operating system. The hardware environment includes Intel Core i5-10400F CPU, Nvidia RTX3060 GPU, and 32GB of memory. In the software environment, Python 3.8 is used as the development language, the deep learning framework is Pytorch 1.10.0, and other related software includes Layui 2.7.5, Openflow 1.3, Openvswitch 2.9.8, Flask 1.1.2, and Mininet-wifi 2.4.3. The datasets used in the experiment are KDD99 dataset and OODS dataset. The KDD99 dataset contains massive network connection records, each labeled as normal or some type of attack, mainly used to evaluate network intrusion detection systems. The OODS dataset comes from an anomaly detection dataset website and covers various application scenarios in different fields. It is commonly applied to study the performance of models in detecting unknown data distributions. After preprocessing, the dataset contains 40384 pieces of data, including 33025 pieces of Normal data, 14435 pieces of DoS data, and 7076 pieces of Replay data. To reflect the uneven distribution of the dataset, random sampling is performed on the abnormal sample DoS and Replay data, and the training and testing sets are divided in an 8:2 ratio. In

the experiment, in order to ensure that the KDD99 and OODS datasets are suitable for training and testing the VAE-GRU-XGBoost model, preprocessing operations were first performed on the data. After conducting routine data cleaning, the data in the dataset was subjected to feature standardization and normalization, as well as label conversion and other operations. Furthermore, due to the highly imbalanced attack types in the KDD99 and OODS datasets, the study oversampled and undersampled the data to achieve a more balanced distribution of positive and negative samples in the training set. In addition, the study will randomly sample the attack signals DoS and Replay data that need to be utilized in the experiment to generate suitable training and testing sets, avoiding bias caused by class imbalance. The traffic types and attack types in the KDD99 dataset are shown in Table 1.

In order to avoid dealing with new attack patterns or inconsistent distributions of training and testing data, the study used a training set that includes various attack patterns to enhance the robustness of the model. In addition, the learning mechanism of XGBoost structure can enable the model to continuously adapt to new attack patterns. When new attack types or patterns emerge, the model will fine tune and update with new data without fully retraining, thereby improving its detection ability for unknown attacks. The specific experimental configuration is shown in Table 2.

## 4.2. Performance verification of vae feature extraction model based on GRU

To verify the VAE on the ground of GRU improvement, the reconstruction error changes of the VAE model before and after GRU improvement are compared, as shown in Fig 7. After the GRU improvement, the reconstruction loss of the VAE decreased faster, and the final converged reconstruction loss value was also smaller than the model before the GRU improvement. When the number of iterations approached 50, the VAE based on GRU improvement gradually converged, and the reconstruction loss value at this time was 0.019. The VAE feature extraction model before GRU improvement only tended to converge after nearly 90 iterations, with a final reconstruction loss value of 0.041. The reconstruction loss of the VAE feature extraction model based on GRU decreased by 53.65% compared with before the improvement, indicating that the VAE feature extraction model based on GRU effectively constrained the reconstruction error.

To further validate the performance of the VAE feature extraction model on the ground of GRU improvement, the feature extraction accuracy of this model is compared with other existing methods, including XGBoost classification model,

**Table 1. Examples of traffic types and attack types in the KDD99 dataset.**

| Category | Description | Attack Types |
| --- | --- | --- |
| Normal Traffic | Regular network traffic without any abnormal behavior | Normal |
| Denial of Service (DoS) | Attacks that aim to make the target service unavailable by sending large amounts of data or exploiting protocol vulnerabilities | Apache2, back, mailbomb, smurf, named, xsnoop, etc. |
| Probe Attack | Used to collect network information or perform system scans | Nmap, ipsweep, saint, mscan, etc. |
| User to Root (U2R) | Attacks that gain unauthorized root access to a system | Ps, buffer_overflow, perl, rootkit, sqlattack, etc. |
| Remote to Local (R2L) | Remote attackers exploit weak passwords or vulnerabilities to gain access to local systems | guess_passwd, ftp_write, imap, phf, warezclient, etc. |

**Table 2. Specific experimental configuration.**

| Hardware environment | Configuration | Software environment | Configuration |
| --- | --- | --- | --- |
| Operating System | Ubuntu 18.04 LTS | The server | Dell_PowerEdge-T440 |
| CPU | Intel Core i5-10400F | Layui | 2.7.5 |
| GPU | Nvidia RTX3060 | Openflow | 1.3 |
| Memory | 32G | Openvswitch | 2.9.8 |
| Language | Python3.8 | flask | 1.1.2 |
| Deep Learning Framework | Pytorch 1.10.0 | Mininet-wifi | 2.4.3 |

SVM, and Generative Adversarial Network (GAN) model [29,30]. The KDD99 and OODS datasets are commonly used public datasets in the field of network intrusion detection. The KDD99 dataset contains real-world scenario data extracted from network traffic, while the OODS dataset is not limited to network traffic attacks, but also covers various anomaly detection tasks in different fields, making it highly representative in detecting unknown intrusion scenarios [31]. Therefore, the KDD99 and OODS datasets can represent real scenes. In order to demonstrate the effectiveness of the experiment, the study also additionally utilized the current new dataset CICIDS 2017 to demonstrate the effectiveness of the experiment. The comparison of feature extraction accuracy among different models is shown in Fig 8. From Fig 8(a), in the KDD99, the extraction accuracy of the VAE optimized by GRU was 94.19%, which was 5.56%, 14.65%, and 16.94% higher than the accuracy of models such as XGBoost, SVM, and GAN, which were 88.63%, 79.54%, and 77.25%, respectively. From Fig 8(b), in the OODS, the extraction accuracy of the VAE optimized by GRU was 95.07%. Compared with the accuracy of 89.22%, 78.17%, and 76.20% of the other three methods, it improved by 5.85%, 16.90%, and 18.87%, respectively. From the above, the VAE feature extraction model based on GRU improvement has high feature extraction accuracy, which is significantly better than other methods. From Fig 8(c), it can be seen that in the CICIDS 2017 dataset, the extraction accuracy of the GRU improved VAE model is 94.28%, which is 6.22%, 16.64%, and 18.45% higher than the accuracy of 88.06%, 77.64%, and 75.83% of the other three methods, respectively. From the above, it can be seen that the VAE feature extraction model based on GRU improvement has high feature extraction accuracy, which is significantly better than other methods. In addition, the studied model performed well on multiple datasets, further demonstrating its high generalization ability and robustness.

To verify the efficiency of the VAE feature extraction model based on GRU, the feature extraction efficiency of different models is compared, as displayed in Table 3. In the KDD99, when the sample size was 10000, the feature extraction time of the VAE model based on GRU was the shortest at 0.030s. Compared with models such as XGBoost, SVM, and GAN, the time was reduced by 97.11%, 97.25%, and 98.80%, respectively. The designed model has significantly higher efficiency than other models, and this trend continues with the increase of sample size. In the OODS dataset, when the sample size was 40000, the feature extraction time of the VAE model based on GRU was the longest at 0.112s. Compared with the other three models, the time was shortened by 98.13%, 98.41%, and 99.03%, respectively. Overall, the VAE feature extraction model based on GRU improvement has significantly better extraction efficiency than other models.

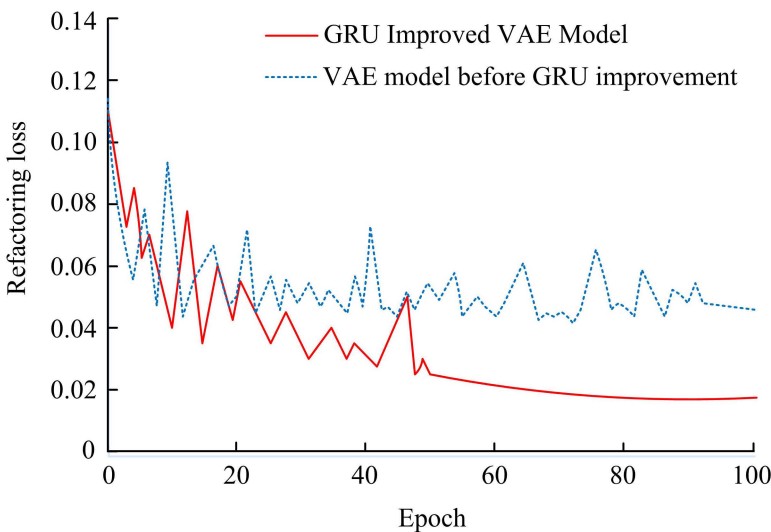

**Fig 7. Comparison of reconstruction error changes in VAE feature extraction models before and after GRU improvement.**

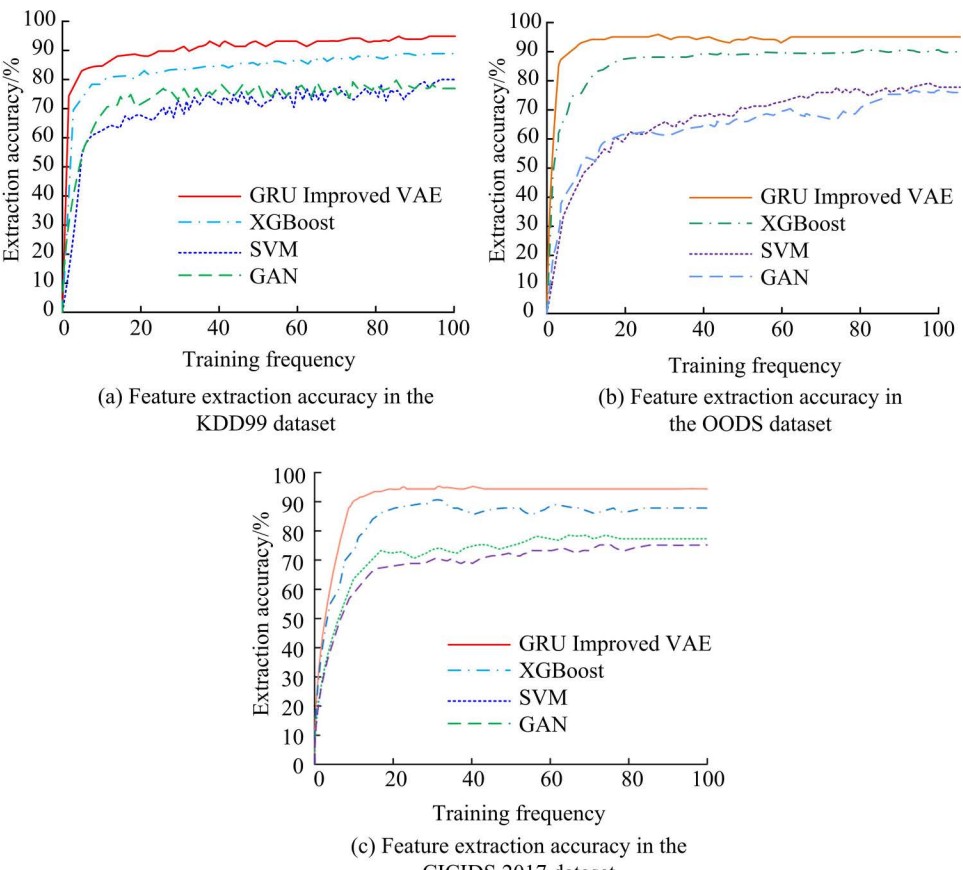

Fig 8. Comparison of feature extraction accuracy among different models.

Table 3. Comparison of feature extraction efficiency among different models.

| Data set | Sample quantity/piece | Time of different models/s | | | |
|---|---|---|---|---|---|
| | | GRU Improved VAE | XGBoost | SVM | GAN |
| KDD99 | 10000 | 0.030 | 1.041 | 1.092 | 2.513 |
| | 20000 | 0.057 | 2.085 | 2.134 | 5.420 |
| | 30000 | 0.092 | 3.149 | 4.284 | 7.984 |
| | 40000 | 0.101 | 5.167 | 6.495 | 10.651 |
| OODS | 10000 | 0.032 | 1.164 | 1.123 | 2.346 |
| | 20000 | 0.060 | 2.649 | 2.264 | 5.566 |
| | 30000 | 0.088 | 4.034 | 4.987 | 8.064 |
| | 40000 | 0.112 | 6.017 | 7.063 | 11.640 |

## 4.3. Performance verification based on VAE-GRU-Xgboost model

To verify the attack detection performance of the VAE-GRU-XGBoost, it is compared with other advanced intrusion detection models. Different traffic sub-classes include Normal traffic, Denial of Service (DoS) attack traffic, Probe attack traffic, Remote to Local (R2L) attack traffic, and User to Root (U2R) attack traffic. The detection rate is displayed in Fig 9. From

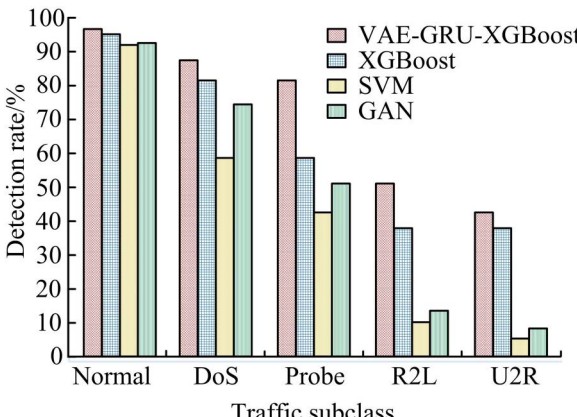

**Fig 9. Comparison of detection rates of different network intrusion detection models.**

Fig 9, the detection rate of the Normal traffic was the highest, while the U2R class was the lowest. This is because there are few attack samples in the dataset for this class, and the model lacks sufficient training. However, the detection rate of different traffic sub-classes on the basis of the VAE-GRU-XGBoost exceeded other methods. In the Normal class, the research model was 96.83%, which was 1.54%, 4.79%, and 4.20% higher than the 95.29%, 92.04%, and 92.63% detection models on the basis of XGBoost, SVM, and GAN, respectively. In the U2R class, the research model was 42.87%, which was 4.96%, 37.37%, and 34.63% higher than the 37.91%, 5.50%, and 8.24% of the other three models, respectively. Overall, the VAE-GRU-XGBoost has excellent attack detection performance.

The studied model has different execution operations when detecting different types of attacks: 1. In response to DoS attacks, the VAE structure can identify sudden high loads or abnormal request frequencies in traffic, capturing potential characteristics of DoS attacks. The GRU structure can identify temporal patterns in DoS attacks and capture these dynamic features through temporal dependencies. Finally, the XGBoost classifier classifies the extracted features, detects abnormal traffic patterns, and marks them as DoS attacks. Personal evaluation result: The accuracy of this model for DoS attacks is usually high, especially in cases where there are obvious features such as sudden changes in traffic patterns and high-frequency requests. The false positive rate is low, and the model is more sensitive to sudden traffic response.

1. When dealing with SQL injection attacks, the VAE structure can identify the characteristics of SQL injection attacks, such as abnormal database query requests, SQL keywords in input data, etc. The GRU structure can capture changes in input through temporal analysis, such as abnormal time intervals in malicious requests or coherent attack sequences. Then use XGBoost to classify and detect the presence of SQL injection patterns in the features. Personal evaluation result: The model performs well in detecting SQL injection attacks and accurately captures malicious input when analyzing HTTP requests and database interactions.

2. When targeting malicious software propagation attacks, the VAE structure can analyze abnormal file transfer characteristics or behavior patterns in network traffic, and identify whether there are signs of malicious software propagation. The GRU structure can help capture temporal patterns of file transfer or command execution, such as malicious software that may rapidly spread in the network and connect through multiple steps, and GRU can identify these temporal dependencies. Finally, XGBoost is used to determine and classify the type it belongs to. Personal evaluation results: The model also performs well in detecting the spread of malicious software, especially when capturing anomalies in network behavior, and can identify time series changes during the propagation process.

To further validate the attack detection performance of the VAE-GRU-XGBoost, a performance comparison analysis is conducted on small sample attack traffic such as R2L and U2R, as shown in Table 4. From Table 4, in R2L class detection, the accuracy, recall, and F1 value of the VAE-GRU-XGBoost reached 96.73%, 93.14%, and 93.56%, respectively. In the detection of U2R class, the accuracy, recall, and F1 of this model reached 97.25%, 92.08%, and 94.41%, respectively, with significantly higher performance indicators than the other three models. In the detection of small sample attack traffic, the attack detection performance based on VAE-GRU-XGBoost model outperforms the other three models.

To verify the classification performance of the VAE-GRU-XGBoost, a confusion matrix experiment is conducted, as shown in Fig 10. From Fig 10(a), in the training set, the VAE-GRU-XGBoost model had classification accuracy greater than 0.91 for different attack traffic samples, with classification accuracy of 0.98, 0.92, 0.94, 0.96, and 0.91 for Normal, DoS, Probe, R2L, and U2R, respectively. From Fig 10(b), in the testing set, the classification accuracy was also relatively high, all exceeding 0.92. The classification accuracy of Normal, DoS, Probe, R2L, and U2R was 0.97, 0.94, 0.92, 0.93, and 0.92, respectively. From the above, the VAE-GRU-XGBoost has high feature extraction and classification performance.

**Table 4. Comparison of detection performance of different detection models for small samples.**

| Small sample attack traffic | Detection model | Performance index | | |
|---|---|---|---|---|
| | | Accuracy/% | Recall/% | F1 value |
| R2L | VAE-GRU-XGBoost | 96.73 | 93.14 | 93.56 |
| | XGBoos | 92.04 | 84.46 | 77.67 |
| | SVM | 87.54 | 78.23 | 66.51 |
| | GAN | 72.16 | 65.07 | 72.65 |
| U2R | VAE-GRU-XGBoost | 97.25 | 92.08 | 94.41 |
| | XGBoos | 91.97 | 85.06 | 78.91 |
| | SVM | 88.90 | 79.04 | 69.42 |
| | GAN | 75.19 | 64.98 | 70.51 |

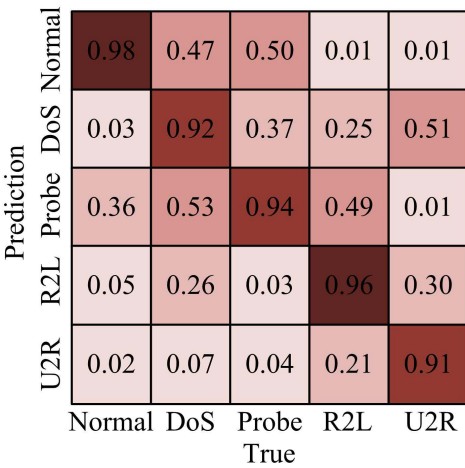
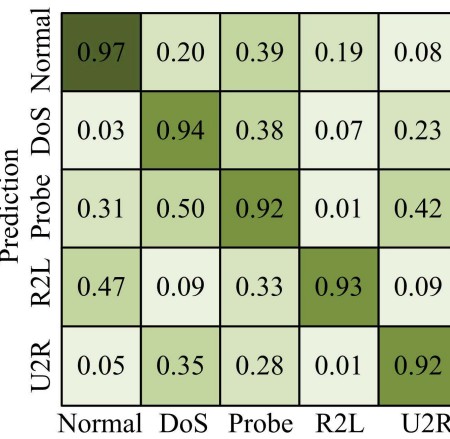

(a) The confusion matrix results in the training set

(b) The confusion matrix results in the test set

**Fig 10. Experimental results of confusion matrix.**

To verify the performance advantages of the research model in real-world scenarios, it was compared and analyzed with advanced network intrusion detection models such as DeepAutoEncoder (DAE), Random Forest (RF), and Convolutional Neural Networks (CNN).In addition, in order to compare the fairness of the methods, the study also added a GRU XGBoost hybrid model for comparative experiments. The real scenario of the experiment refers to network intrusion detection conducted in actual network environments, where the model is applied to various network traffic data in the real world, rather than just testing on standard datasets or laboratory environments. The performance comparison of different network intrusion detection models is shown in Table 5. From Table 5, it can be seen that the research model outperforms other network intrusion detection models in terms of accuracy, speed, and robustness. From the perspective of robustness, the studied model has high applicability, while DAE only depends on the type of attack, RF is affected by data inconsistency, and CNN requires a large amount of data training. In terms of accuracy, although the accuracy of each model is higher than 90%, the studied model has a higher accuracy of 97.58% and better performance. From the perspective of model speed, the training time and inference time of the research model are the shortest, at 1.15 hours and 0.003 seconds, respectively. While the LSTM XGBoost model performs well in handling various types of attacks, the proposed model offers better adaptability and generalization capabilities.However, the model presented in this article demonstrates higher adaptability and generalization ability when dealing with complex features, thanks to its VAE feature extraction mechanism. In addition, from the perspective of inference time, the LSTM XGBoost model has a longer inference time, indicating that it is not suitable for most real-time detection needs. Therefore, it can be concluded that the research model outperforms other models in terms of accuracy, speed, and robustness in real-world scenarios.

To further validate the performance of the network intrusion detection model on the basis of VAE-GRU-XGBoost, the Receiver Operating Characteristic (ROC) curves of this model were compared with other models, as shown in Fig 11. From Fig 11(a), in the KDD99 dataset, the AUC value of the VAE-GRU-XGBoost model reached 97.48%, which was 11.81%, 12.92%, and 18.41% higher than that of the XGBoost, SVM, and GAN models, respectively. From Fig 11(b), in the OODS dataset, the AUC value of the VAE-GRU-XGBoost model reached 95.24%, which was 6.77%, 10.19%, and 30.76% higher than the AUC of the other three models, respectively. From the above, the VAE-GRU-XGBoost has high intrusion detection accuracy.

To further validate the role of each structure in the VAE-GRU-XGBoost, ablation experiments are carried out, as displayed in Table 6. "√" indicates the existence of the module, and "/" indicates its non-existence. From Table 6, when using only the XGBoost classifier, the precision, recall, and F1 of the model were 77.64%, 78.36%, and 78.01%, respectively. The precision, recall, and F1 of the VAE optimized by GRU were 91.05%, 90.49%, and 92.84%, respectively. This indicates that when only the XGBoost module is used without VAE and GRU, the model performance significantly decreases, indicating that using XGBoost alone is difficult to capture the complex features of data. The precision, recall, and F1 of the VAE-GRU-XGBoost reached 95.86%, 96.69%, and 97.15%, demonstrating the best performance in network intrusion detection tasks. This indicates that VAE and GRU modules play an important role in feature extraction and time series modeling. From the above, the VAE-GRU-XGBoost has a high detection performance, indicating that each module in the model contributes to the final performance and plays a synergistic role in the model.

**Table 5. Performance comparison of different network intrusion detection models.**

| Model | Accuracy/% | Training time/hour | Inference time/s | Robustness |
|---|---|---|---|---|
| VAE-GRU-XGBoost | 97.58 | 1.15 | 0.003 | High adaptability |
| DAE | 93.08 | 10.51 | 0.167 | Affected by inconsistent data |
| RF | 92.06 | 3.46 | 1.644 | Not adapted to new types of attacks |
| CNN | 91.68 | 14.67 | 0.076 | More data training is needed |
| LSTM-XGBoost | 95.32 | 2.25 | 0.052 | Good adaptability with moderate data |

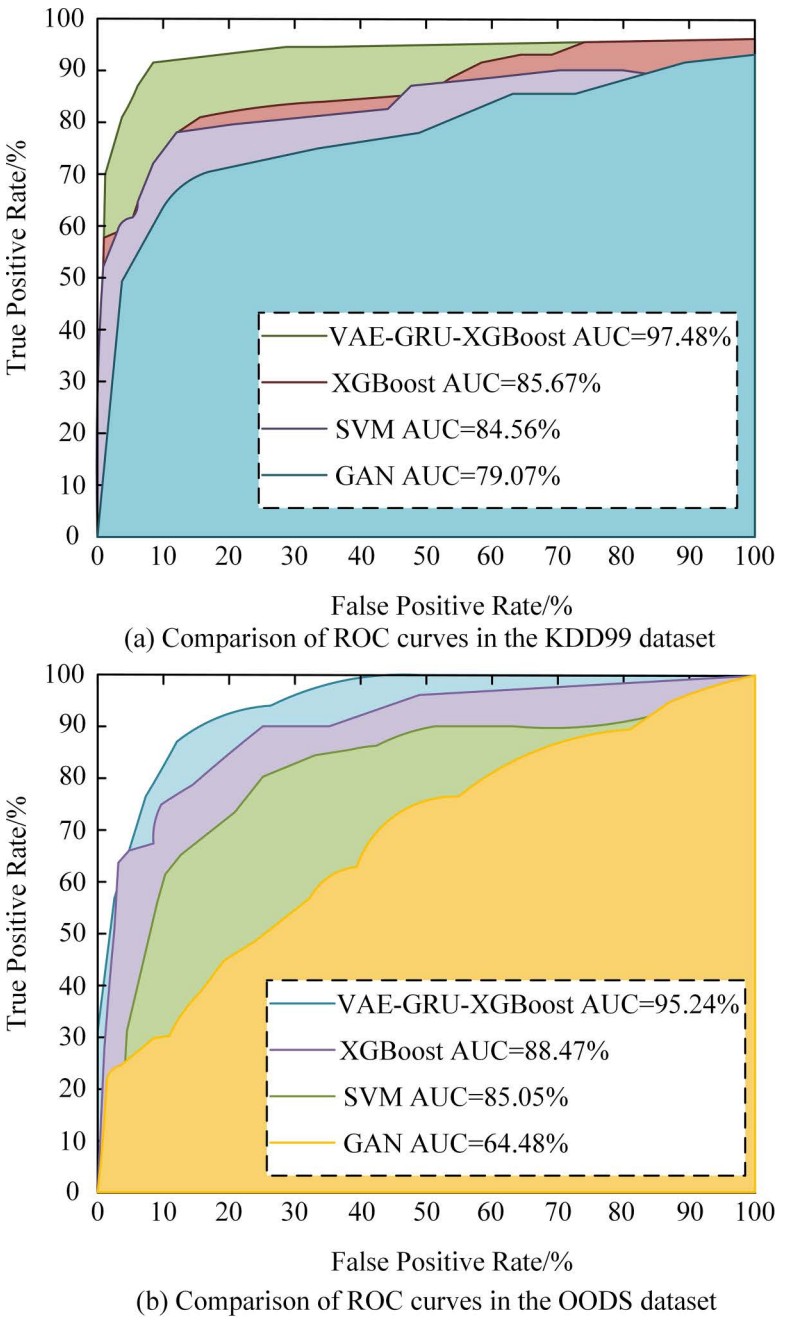

(a) Comparison of ROC curves in the KDD99 dataset

(b) Comparison of ROC curves in the OODS dataset

**Fig 11. ROC curve comparison.**

In order to verify the impact of different feature addition orders on model performance, a series of experiments were conducted by randomly arranging input features, and the feature addition order and corresponding AUC values of each experiment were recorded. The experimental results of different feature addition orders are shown in Table 7. From Table 7, it can be seen that the AUC values after modifying the order are all greater than 0.80. However, changing the order of feature addition does indeed affect the performance of the model. When adding features of lower importance first, it can lead

**Table 6. Ablation experiment.**

| Method | VAE | GRU | XGBoost | Precision/% | Recall/% | F1 value/% |
|---|---|---|---|---|---|---|
| XGBoost | / | / | √ | 77.64 | 78.36 | 78.01 |
| GRU Improved VAE | √ | √ | / | 91.05 | 90.49 | 92.84 |
| VAE-GRU-XGBoost | √ | √ | √ | 95.86 | 96.69 | 97.15 |

**Table 7. Experimental results of different feature addition orders.**

| Experiment number | Feature addition order | AUC Value | Above Threshold | Result Validity |
|---|---|---|---|---|
| 1 | src_bytes→protocol_type→hot→num_failed_logins→diff_srv_rate | 0.85 | Yes | Valid |
| 2 | protocol_type→hot→num_failed_logins→diff_srv_rate→src_bytes | 0.83 | No | Invalid |
| 3 | hot→src_bytes→num_failed_logins→diff_srv_rate→protocol_type | 0.84 | Yes | Valid |
| 4 | diff_srv_rate→num_failed_logins→src_bytes→protocol_type→hot | 0.86 | Yes | Valid |
| 5 | num_failed_logins→diff_srv_rate→src_bytes→protocol_type→hot | 0.82 | No | Invalid |
| 6 | flag→dst_host_same_srv_rate→srv_count→dst_host_same_src_port_rate | 0.80 | No | Invalid |
| 7 | dst_host_same_srv_rate→srv_count→dst_host_same_src_port_rate→flag | 0.81 | No | Invalid |
| 8 | srv_count→dst_host_same_src_port_rate→flag→dst_host_same_srv_rate | 0.83 | Yes | Valid |
| 9 | dst_host_same_src_port_rate→srv_count→flag→dst_host_same_srv_rate | 0.84 | Yes | Valid |
| 10 | dst_host_same_srv_rate→dst_host_rerror_rate→service→dst_bytes | 0.87 | Yes | Valid |

to a decrease in AUC value. If the AUC is lower than the set threshold, it will have a negative impact on the learning performance of the model, thereby affecting the effectiveness of the results.

## Conclusion

As artificial intelligence continues to evolve, anomaly detection technology is widely used in security monitoring, fault detection, and network security fields. To improve the accuracy of network intrusion detection, a network intrusion detection model on the basis of VAE-GRU-XGBoost was constructed by fully integrating the advantages of modules such as VAE, XGBoost, and GRU. The VAE feature extraction model ground on GRU improvement reduced the reconstruction loss value to 0.019, which was 53.65% lower than before the improvement, indicating that the improved model effectively constrained the reconstruction error. The extraction accuracy of the VAE model optimized by GRU reached 94.19% in the KDD99 dataset and 95.07% in the OODS dataset, significantly higher than other feature extraction models. As for efficiency and performance, when the sample size was 10000 in the KDD99 dataset, the had the shortest feature extraction time of 0.030s, which was 97.11%, 97.25%, and 98.80% lower than the feature extraction time of XGBoost, SVM, and GAN models, respectively. In the OODS dataset, when the sample size was 40000, the longest feature extraction time was 0.112s, which was 98.13%, 98.41%, and 99.03% shorter than the feature extraction time of the other three models, respectively. In R2L class detection, the accuracy, recall, and F1 of the VAE-GRU-XGBoost reached 96.73%, 93.14%, and 93.56%, respectively. In U2R class detection, the accuracy, recall, and F1 are 97.25%, 92.08%, and 94.41%, respectively. This indicates that the model significantly outperformed other models in small sample attack traffic detection performance. In the confusion matrix experiment, the model achieved classification accuracy greater than 0.91 for different attack traffic samples in both the training and testing sets, indicating its high feature extraction and classification performance. From the ROC comparison experiment, the AUC value of this model was as high as 97.48% in the KDD99 dataset and 95.24%

in the OODS dataset, demonstrating excellent intrusion detection performance. From the ablation experiment, each module in the model plays a synergistic role, comprehensively improving the performance. In summary, the VAE-GRU-XGBoost network intrusion detection model for network security has effectively improved the intrusion detection accuracy. However, the actual methods of network attacks are complex and varied, and the research has only explored common attack traffic such as R2L and U2R. The research results are not comprehensive enough. Future research can analyze rare attack traffic and provide more comprehensive and reliable technical support for network security protection.

## Supporting information

**S1 File. Minimal data set definition.**
(DOC)

## Author contributions

**Formal analysis:** Xiaohong Zheng.

**Investigation:** Yu Chen.

**Methodology:** Yu Chen.

**Project administration:** Nan Wang.

**Software:** Xiaohong Zheng.

**Supervision:** Nan Wang.

**Writing – original draft:** Yu Chen.

**Writing – review & editing:** Xiaohong Zheng, Nan Wang.

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
