## [Decision Letter · Decision Letter 0]

PONE-D-25-07820Construction of VAE-GRU-XGBoost Intrusion Detection Model for Network SecurityPLOS ONE

Dear Dr. Zheng,

Thank you for submitting your manuscript to PLOS ONE. After careful consideration, we feel that it has merit but does not fully meet PLOS ONE’s publication criteria as it currently stands. Therefore, we invite you to submit a revised version of the manuscript that addresses the points raised during the review process.

We look forward to receiving your revised manuscript.

Kind regards,

Guangyin Jin

Academic Editor

PLOS ONE

Reviewers' comments:

Reviewer's Responses to Questions

**Comments to the Author**

1. Is the manuscript technically sound, and do the data support the conclusions?

Reviewer #1: Partly

2. Has the statistical analysis been performed appropriately and rigorously? 

Reviewer #1: N/A

3. Have the authors made all data underlying the findings in their manuscript fully available?

Reviewer #1: No

4. Is the manuscript presented in an intelligible fashion and written in standard English?

Reviewer #1: Yes

5. Review Comments to the Author

Reviewer #1: Overview

This manuscript proposes a novel hybrid model for network intrusion detection that combines VAE, GRU, and XGBoost. Their model shows great performance at feature extraction and classification of network traffic to detect intrusions on the KDD99 and OODS datasets, as well as efficient processing times. The authors also include detailed literature review, discussions and ablation study.

Major Issues

1. Insufficient Technical Details: The paper lacks a precise mathematical explanation of how VAE, GRU, and XGBoost components are integrated. While individual components are explained, their interconnection isn't clearly articulated. Implementation details such as hyperparameters, training strategies, and optimization techniques are missing for the proposed methods, as well as compared methods like SVM, GAN and CNN.

Also, some of the math equations seem vague. For example, Equation (1) and (2) seem to imply that both the encoder and the decoder contain just one hidden layer respectively. There is also no clear label of the shape of input, parameters and output.

The feature selection accuracy mentioned around Figure 8 is not clearly defined.

2. Limited Novelty and Design Justification: The authors claim novelty in combining these three techniques, but don't sufficiently explain why this particular combination was chosen over alternatives. Also in the related work section, the authors mentioned that “Cao B et al. combined deep learning technology and GRU design to improve the intrusion detection accuracy.” It’s straightforward to see the close relationship between Cao B et al.’s model and the proposed one. However, this is no discussion regarding this matter.

Also, Cao B et al.’s model can be trained end-to-end as they used a softmax layer after all feature extraction steps, while the authors use XGBoost which I don’t think can be applied end to end. I would like to see more clarification/discussion on this choice.

The feature selection process discussed around Figure 4 needs to be justified as well. The described process could be highly dependent on the order of added features. It would be good to see if different order gives similar performance.

3. Unfair Comparisons: The compared methods like standard XGBoost, SVM and GAN are relatively simple and naïve approaches, compared to the proposed method. So even if the proposed method outperforms others, there’s not enough evidence to make the intended conclusion. At least similar hybrid models should be compared.

4. Vague Math Equations: Some of the math equations seem vague. For example, Equation (1) and (2) seem to imply that both the encoder and the decoder contain just one hidden layer respectively. There is also no clear label of the shape of input, parameters and output.

Minor Issues

1. Inadequate Dataset Description: There are no examples of data from KDD99 and OODS datasets. It would give the readers more context if some examples can be shown at least in the SI.

2. Typos: In the abstract, "The results *showed* *demonstrated*..." Table 2 and Figure 11 show “XGBoos” instead of “XGBoost.”

6. PLOS authors have the option to publish the peer review history of their article (what does this mean? ). If published, this will include your full peer review and any attached files.

**Do you want your identity to be public for this peer review?** For information about this choice, including consent withdrawal, please see our Privacy Policy .

Reviewer #1: No

---

## [Author Response · Author response to Decision Letter 1]

14 May 2025

The manuscript has been improved according to comments.

---

## [Decision Letter · Decision Letter 1]

Construction of VAE-GRU-XGBoost Intrusion Detection Model for Network Security

PONE-D-25-07820R1

Dear Dr. Zheng,

We’re pleased to inform you that your manuscript has been judged scientifically suitable for publication and will be formally accepted for publication once it meets all outstanding technical requirements.

Kind regards,

Guangyin Jin

Academic Editor

PLOS ONE

Additional Editor Comments (optional):

Reviewers' comments:

Reviewer's Responses to Questions

**Comments to the Author**

1. If the authors have adequately addressed your comments raised in a previous round of review and you feel that this manuscript is now acceptable for publication, you may indicate that here to bypass the “Comments to the Author” section, enter your conflict of interest statement in the “Confidential to Editor” section, and submit your "Accept" recommendation.

Reviewer #1: All comments have been addressed

2. Is the manuscript technically sound, and do the data support the conclusions?

Reviewer #1: Yes

3. Has the statistical analysis been performed appropriately and rigorously? 

Reviewer #1: N/A

4. Have the authors made all data underlying the findings in their manuscript fully available?

Reviewer #1: Yes

5. Is the manuscript presented in an intelligible fashion and written in standard English?

Reviewer #1: Yes

6. Review Comments to the Author

Reviewer #1: All comments have been addressed. The new experiment results make the conclusion more solid and complete the story.

7. PLOS authors have the option to publish the peer review history of their article (what does this mean? ). If published, this will include your full peer review and any attached files.

**Do you want your identity to be public for this peer review?** For information about this choice, including consent withdrawal, please see our Privacy Policy .

Reviewer #1: No

---

## [Editor Report · Acceptance letter]

PONE-D-25-07820R1

PLOS ONE

Dear Dr. Zheng,

I'm pleased to inform you that your manuscript has been deemed suitable for publication in PLOS ONE. Congratulations! Your manuscript is now being handed over to our production team.

Kind regards,

on behalf of

Dr. Guangyin Jin

Academic Editor

PLOS ONE